# The Key Role of Epithelial to Mesenchymal Transition (EMT) in Hypertensive Kidney Disease

**DOI:** 10.3390/ijms20143567

**Published:** 2019-07-21

**Authors:** Teresa Seccia, Brasilina Caroccia, Maria Piazza, Gian Paolo Rossi

**Affiliations:** Hypertension Unit, Department of Medicine-DIMED, University Hospital, 35128 Padova, Italy

**Keywords:** epithelial to mesenchymal transition (EMT), hypertension, angiotensin II, endothelin

## Abstract

Accumulating evidence indicates that epithelial-to-mesenchymal transition (EMT), originally described as a key process for organ development and metastasis budding in cancer, plays a key role in the development of renal fibrosis in several diseases, including hypertensive nephroangiosclerosis. We herein reviewed the concept of EMT and its role in renal diseases, with particular focus on hypertensive kidney disease, the second leading cause of end-stage renal disease after diabetes mellitus. After discussing the pathophysiology of hypertensive nephropathy, the ‘classic’ view of hypertensive nephrosclerosis entailing hyalinization, and sclerosis of interlobular and afferent arterioles, we examined the changes occurring in the glomerulus and tubulo-interstitium and the studies that investigated the role of EMT and its molecular mechanisms in hypertensive kidney disease. Finally, we examined the reasons why some studies failed to provide solid evidence for renal EMT in hypertension.

## 1. Introduction

More than 20 million people have chronic kidney disease (CKD) in the US, with 3% of them developing end-stage renal disease (ESRD) (https://www.usrds.org/adr.aspx), which accounts for 7% of Medicare expenditures. In the EU, the last annual report of the European Renal Association—European Dialysis and Transplant Association (ERA-EDTA) Registry indicated that about 500 thousand people were on renal replacement therapy in 2018 [1]. Moreover, the medical spending is held to be much greater, because ESRD patients are at high risk of cardio- and cerebrovascular vascular disease, which is projected to rise even more, owing to population ageing in the next decades [2]. Arterial hypertension (HT), with a prevalence of 30–45% in the adult population, is the major cardiovascular risk factor and the second cause of ESRD after diabetes mellitus [3,4]. However, the relationship between the kidney and HT is double-threaded, in that nephropathies usually cause HT and high blood pressure damages the kidney [5].

Even though classically described as nephroangiosclerosis and hyalinosis of the glomerular tuft [6,7], hypertensive nephropathy has been more recently found to involve not only the glomerular and vascular compartments, but also the interstitium, because of the development of tubular-interstitial fibrosis (TIF) and, ultimately, ESRD [8,9].

Depending on the underlying pathophysiology and/or severity of HT, multiple mechanisms can concur to induce TIF, among which is epithelial-to-mesenchymal transition (EMT), a process originally described in organ development and metastasis budding in cancer, and later observed in several diseases [10,11]. Although potentially capable of affecting any injured epithelial tissue, whether and to what extent EMT plays role in the kidney in hypertensive nephropathy remains unclear [12,13,14]. Hence, we shall herein focus on the EMT process and its involvement in hypertensive nephropathy by exploiting a methodology to systematically review the English-written literature by means of a PICO (Patient or Problem, Intervention, Comparison, Outcome) search strategy (Table 1) with selection of the articles based on their relevance, rigorousness of study design and methodology, and appropriateness of data interpretation.

## 2. Tubulointerstitial Damage and Fibrosis in Hypertension

In 1993, an involvement of tubule-interstitium in the hypertensive disease was postulated for the first time by Fine [15], who put forward the ‘chronic hypoxia hypothesis’. According to this contention, glomerular injury, primarily determined by HT, would induce hypoxia and endothelial damage of peritubular capillaries [15,16], thus stimulating tubular inflammation and fibrosis. Although appealing, this hypothesis remained unproven until one study showed peritubular capillaries rarefaction and tubulointerstitial scarring in human biopsies of CKD hypertensive patients [17] and models of hypertensive disease documented an involvement of the interstitium in HT [8,9,18,19].

Fibroblasts are strategically posed in the interstitium to sense circulating toxins or factors and environmental changes and can be activated by local tissue injury to differentiate into myofibroblasts that synthesize extracellular matrix deposition and collagens. However, fibroblasts can also act as inflammatory effector cells: They express immune receptors as Toll-like receptors (TLRs) and activate NF-κB signaling with ensuing release of cytokines and chemokines that amplify kidney damage. In the hypertensive nephropathy, TIF usually associates with glomerular damage, loss of nephrons, and decline in renal function. Nephrons are then focally replaced by fibrotic tissue along a scarring process, finally leading to ESRD [20]. Thus, chronic and progressive tubular damage invariably leads to TIF.

Several factors can induce TIF. One of them is angiotensin II (Ang II), a major effector of the renin angiotensin aldosterone system (RAAS), which acts as a potent vasoconstricting and pro-inflammatory factor. Using a transgenic renin-dependent model of severe HT and cardiovascular and renal damage created with insertion of the mouse renin gene into the rat genome, the TG(mRen2)27 rat, we found that a blockade of the Ang II type 1 (AT1) receptor with irbesartan prevented TIF, thus supporting a fibrogenic role of Ang II [8]. Interestingly, in this model, TIF was also prevented by bosentan, a mixed ET_A_-ET_B_ endothelin receptor antagonist, but worsened by BMS-182874, a selective ET_A_ receptor antagonist, suggesting a key role of endothelin-1 (ET-1) acting via the ET_B_ receptor subtype in triggering renal fibrosis [8].

It is important to note that dual inhibition of a angiotensin-converting enzyme (ACE) and a neutral endopeptidase also prevented TIF; moreover, this favorable effect was abolished by the bradykinin B2 receptor antagonist icatibant, thus implicating the B2 subtype receptor in counterbalancing the deleterious effects of Ang II in the tubule-interstitium [9].

Ang II can promote TIF not only by activating resident fibroblasts, but also by inducing proteinuria or EMT (see later *“EMT in Hypertensive Nephropathy”*). When hypertensive nephropathy is associated with proteinuria, filtered proteins are held to exert a further toxic effect on proximal tubular cells, which can amplify inflammation [21]. Pro-inflammatory factors, such as monocyte chemoattractant protein-1 (MCP-1/CCL2), regulated upon activation normal T-cell expressed and secreted/Chemokine (C-C motif) ligand 5 (RANTES/ CCL5), and fractalkine/CX3CL1 are chemoattractant for monocytes/macrophages and T lymphocytes, which stimulate synthesis of transforming growth factor β1 (TGFβ1), a potent fibroblast activator [14].

## 3. Epithelial-to-Mesenchymal Transition (EMT)

Renal epithelial cells arise during embryogenesis by mesenchymal-to-epithelial transition (MET) [22,23]. For decades the dogma was that epithelial cells maintain its phenotype until death, but, after the seminal study by Gros in 2014 [24], the concept that epithelial cells can switch back to a mesenchymal phenotype via epithelial-to-mesenchymal transition (EMT) was promptly accepted [25,26]. First described in embryogenesis [24], EMT was then observed in inflammation [27], fibrosis [28,29,30], wound healing [31], and cancer progression [32,33,34].

Epithelia are characterized by apical-basal polarity and tight cell–cell junctions between neighboring cells, accounting for cell integrity and stability, whereas mesenchymal and interstitial tissues are formed by elongated and spindle-shaped cells, known as fibroblast-like cells, which have loose cell–cell interactions. Because of their mobility, these cells can migrate and create an anchorage distantly.

The cells undergoing EMT progressively lose epithelial markers, such as E-cadherin, which is the main component of adherent junctions, and acquire markers of mesenchymal phenotype, such as α smooth muscle actin (αSMA) [35]. Loss of cell junctions allows epithelial cells to translate into loss of adhesion, change morphology, and loss apical-basal polarity. After synthesis of matrix metalloproteinases (MMPs), cells cleave and invade the basal lamina to move towards the interstitial space. Migration is driven by newly formed matrix of fibronectin and type I collagen. Upon completion of the epithelial-to-mesenchymal phenotype switch, the cells acquire a myofibroblast phenotype synthesizing αSMA and matrix proteins, the major constituents of TIF [35].

A number of epithelial and mesenchymal markers have been used to track EMT, and a lot of factors have been identified as drivers of EMT. Not unexpectedly, most factors were also found to be involved in embryogenesis, tissue repair, and cancer, suggesting parallels between normal development and malignant growth. TGFβ, a driver of EMT in cancer, is also the most potent factor triggering EMT in the renal cells. After the binding of TGF-β to receptor II TβR-II, TGF-β-bound TβR-II phosphorylates type I receptor (TβR-I) leads to the activation of Smad 2 and 3. Activated Smad2/3 can form a complex with Smad4, which translocates into the nucleus, where it interacts with transcriptional factors and transcriptional co-activators, ultimately inducing synthesis of mesenchymal proteins [36,37,38].

## 4. Markers of EMT

In 1995, Strutz et al. detected S100A4 in tubular cells with persistent inflammation. Since S100A4 was considered a typical protein of mesenchymal cells and only seldom found in tubular cells of healthy kidneys, S100A4 was renamed fibroblast-specific protein–1 (FSP-1) and proposed to be a marker of cells undergoing EMT [39]. However, other investigators were not able to localize FSP-1/S100A4 in activated fibroblasts, and found this protein in immune cells and occasionally in endothelial cells. Hence, the specificity of FSP-1/S100A4 as a marker of EMT was challenged [40,41].

Other markers of EMT have later been proposed. Vimentin is expressed in mesenchymal cells, but also in injured tubular cells, and/or regenerating cells [42,43,44], which led some investigators to consider it as a marker of EMT and others to propose it as a marker of tubular regenerating activity.

αSMA is characteristically expressed in mesenchymal cells, such as vascular smooth muscle cells, pericytes, and interstitial cells around injured tubules and myofibroblasts [45,46]. In contrast, proximal tubular cells contain F-actin, a different isoform that constitutes cytoskeleton microfilaments and cannot bind antibodies against αSMA. However, detection of αSMA cannot be used as a marker specific of EMT, because it cannot discriminate between mesenchymal cells physiologically present in the kidney or myofibroblasts.

The decrease or loss of markers of epithelial cells, such as E-cadherin or ZO-1, is strongly suggestive of EMT, but could also denote tubular cell damage with no transition into mesenchymal cells.

Collagens and proteins that participate to collagen synthesis, such as the collagen-specific molecular chaperone HSP 47 and prolyl-4-hydroxylase, have been also used as markers [47].

Invasion of the basal membrane and cell migration are markers of fully completed EMT. Hence, different markers of EMT have been suggested, with some markers (loss of E-cadherin and ZO-1, synthesis of αSMA and vimentin) indicating the early phenotypic cell switch and others (collagens, matrix invasion) hinting the full transformation of epithelial cells into mesenchymal cells. However, because of the lack of specific markers of transforming cells, *per se*, no marker can provide an unambiguous demonstration of EMT. As discussed below, EMT is a dynamic and transient process that is difficult to visualize. Only an integrated approach evaluating early and late steps can allow identification of EMT.

## 5. EMT as Mediator of TIF

Using a mouse model of severe renal fibrosis, i.e., unilateral ureteral obstruction, Yang and Liu first described EMT in the kidney as an orchestrated process consisting of the classical four steps, i.e., loss of epithelial cell adhesion, de novo synthesis of αSMA, disruption of tubular basement membrane, and cell migration and invasion [48]. In the same model, Iwano et al. found that up to 36% of interstitial matrix-producing cells derived from tubular cells through EMT [35]. However, using models characterized by less pronounced TIF, such as overload proteinuria [48], other investigators found that the contribution of EMT was very scant, and, Humphreys et al. [49], using the model of unilateral ureteral obstruction, were unable to reproduce data from Iwano [35], suggesting that, although renal epithelial cells can acquire mesenchymal markers in vitro, they do not contribute to interstitial myofibroblast cells in vivo. Thus, whether EMT actually contributes to TIF in vivo in all types of nephropathies remains controversial [50,51].

According to a popular view, EMT would be a process by which epithelial cells under stress conditions escape from the unfriendly microenvironment [52]. After an injury, tubular epithelial cells express genes, such as Wnt, Notch, and Hedgehog, which confer resistance to apoptosis and are involved in normal development [52]. However, during physiologic development differentiated epithelial cells proliferate and in an injured kidney epithelial cells acquire partial or full mesenchymal characteristics, thereby preventing proliferation and replacement of damaged cells. Hence, tubular epithelial cells are not locked in a differentiated state, but are ‘plastic cells’ that transform into a different phenotype, but are unable to reach terminal differentiation [52].

Two mechanisms have been proposed: (1) Differentiation of epithelial cells needs “turning off” of Wnt and Notch genes, a process that does not happen in damaged cells; (2) the over-expression of transcription factors, such as Twist, Zeb1, Zeb2, Snai1, and Snai2 favors, mesenchymal reprogramming, preventing terminal differentiation. Elegant studies by Kalluri’s and Nieto’s groups support such contentions: mice over-expressing Snai1 exhibited epithelial plasticity, myofibroblast accumulation, and inflammation, whereas an ablation of Twist and Snai1 induced expression of markers of differentiated cells, such as aquaporins and solute transporters [29,30].

In contrast to other strategies used by epithelial cells to avoid threatening conditions, such as apoptosis or necrosis, EMT is hardly identifiable because of the difficulty of catching exactly the time-point when cells transform into mesenchymal cells and pass the basement membrane [51]. Whether EMT is too rapid to be detected or visualization of EMT needs an integrated methodological approach remains a matter of debate.

## 6. EMT in Hypertensive Nephropathy

Several in vitro studies showed that some factors that induce HT, fibrosis, or both can also drive EMT, suggesting that EMT is a common mechanism underlying TIF [13,53,54,55]. However, only a few studies provided evidences of EMT in vivo in hypertensive nephropathy [12,56,57].

Ang II was found to induce EMT in cell culture models [13,53,54,55,58]. In the rat proximal tubular cell line (NRK52-E), EMT was identified as a morphological change of the epithelial cells from the typical cobblestone pattern to elongated, spindle-shaped mesenchymal cells, and also as a reduction in E-cadherin and synthesis of αSMA protein expression [54]. In a rat model of accelerated ESRD induced by a 14-day infusion of Ang II after preinjury by injection of Habu venom, Falkner et al. observed a progressive increase in αSMA staining, starting from 48 h. However, peritubular interstitial myofibroblasts were also detected after 7 days of Ang II alone infusion, confirming capability of Ang II of inducing EMT [50]. It is important to note that a blockade of the AT1 receptor prevented EMT in models of HT, caused by partial nephrectomy [56] or unilateral ureteral obstruction [57], supporting a role of Ang II in EMT.

As mentioned above, we recently provided evidence that ET-1 can also induce EMT in the kidney [12]: In the same kidney sections of TG(mRen2)27 rats, where we observed TIF, we found a decrease in the epithelial marker E-cadherin along with an increase in the mesenchymal markers αSMA and S100A4, i.e., phenotypic changes that indicated the occurrence of EMT. We also demonstrated co-expression of the epithelial with mesenchymal markers, thus strongly supporting EMT as the mechanism underlying fibrosis in this Ang-II-dependent model of severe hypertension [12].

Moreover, in vitro experiments confirmed that ET-1 can drive EMT in the proximal tubular cells [12]. Exposure of HK2 cells to ET-1 caused disruption of cell junctions and synthesis of αSMA, causing blunting of E-cadherin along with the increase of mesenchymal markers, synthesis of MMP-9, and cell migration [12]. All steps of EMT, such as TIF involved the ET_B_ receptor subtype, which is the predominant receptor subtype in the kidney tubular cells (Figure 1) [12].

## 7. Conclusions

In summary, EMT entails a dynamic sequential process characterized by progressive loss of epithelial markers and acquirement of a mesenchymal phenotype marker that is key to several physiological and pathological processes that need cell plasticity. Although evidences obtained in a model of severe Ang II-dependent hypertension strongly support the contention that EMT contributes to TIF, which is the major determinant of ESRD, likely via the activation of the AT1 and ET_B_ receptors, it remains debated if EMT contributes to TIF in other forms of nephropathy, which are characterized by less prominent TIF.

## Figures and Tables

**Figure 1 ijms-20-03567-f001:**
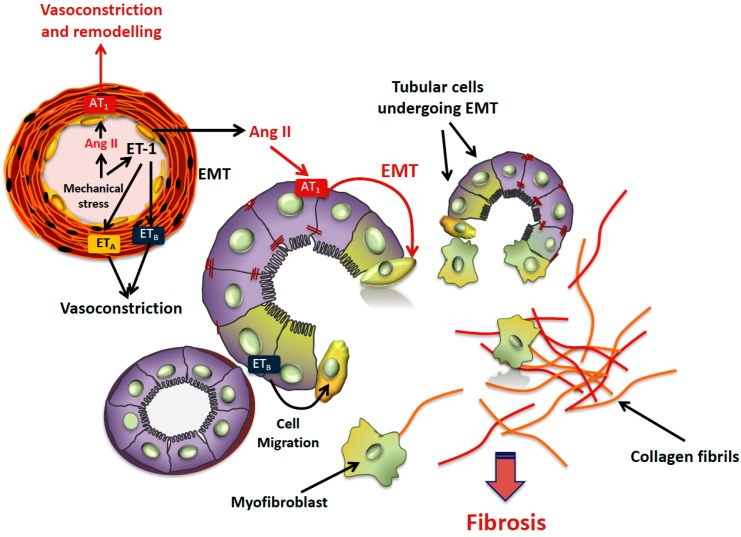
Development of tubule-interstitial damage in the hypertensive disease. When angiotensin II and/or endothelin-1 are abnormally produced, such as under conditions characterized by excess mechanical or oxidative stress, epithelial-to-mesenchymal transition (EMT) is triggered, leading to transformation of tubular cells into myofibroblasts that produce collagens. The final event is tubulo-interstitial fibrosis. Angiotensin II (Ang II), by binding Ang II type 1 (AT1) receptors in the vascular smooth muscle cells, also favors vasoconstriction and vascular remodeling, worsening kidney damage. Vasoconstriction is also mediated by endothelin-1 via ET_A_ and ET_B_ receptors, located at the vascular smooth muscle cells.

**Table 1 ijms-20-03567-t001:** PICO strategy used for the literature search.

Kew Word	Operator	Kew Word	Operator	Kew Word
Hypertension	OR	High blood pressure		
		AND		
Nephropathy	OR	Tubular damage	OR	Tubulointerstitial fibrosis
		AND		
Epithelial-to-mesenchymal transition	OR	EMT	OR	Epithelial-to-mesenchymal transdifferentiation

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
