# Peer review of "The Key Role of Epithelial to Mesenchymal Transition (EMT) in Hypertensive Kidney Disease"

_ijms, 2019, doi:10.3390/ijms20143567_

Reviewer 1 Report

In this manuscript Seccia et al. aim to review the role of EMT in hypertensive kidney disease. They state in the introduction that role of tubular interstium fibrosis in end-stage renal disease has been recently recognized, but the extent of EMT contribution to TIF remains unclear, and they propose to focus on the EMT involvement in Hypertensive nephropathy.

The flow of the paragraphs is not very linear and the titles do not always reflect the content. For example, paragraph 2 is more about Tubolointerstitial Fibrosis in Hypertension rather than generic Tubolointerstitial damage. Two very generic paragraph follow, about EMT and markers of EMT. I’m not sure these are actually needed in this form, given that the review will be part of a special issue on EMT. The paragraph “EMT as a mediator of TIF” is crucial to the review, it should be expanded possibly better explaining the dualism between EMT and epithelial cell plasticity (see Huang S at al Trends Mol Med 2016). Finally, an explanation of the clinical/biological implication of these observations should be included either in the last paragraph “EMT in Hypertensive Neuropathy” or in the conclusions.

Author Response

Manuscript ID: ijms-520214

The Key Role of Epithelial to Mesenchymal Transition (EMT) in Hypertensive Kidney Disease

 Referee 1

In this manuscript Seccia et al. aim to review the role of EMT in hypertensive kidney disease. They state in the introduction that role of tubular interstium fibrosis in end-stage renal disease has been recently recognized, but the extent of EMT contribution to TIF remains unclear, and they propose to focus on the EMT involvement in Hypertensive nephropathy.

The flow of the paragraphs is not very linear and the titles do not always reflect the content. For example, paragraph 2 is more about Tubolointerstitial Fibrosis in Hypertension rather than generic Tubolointerstitial damage. Two very generic paragraph follow, about EMT and markers of EMT. I’m not sure these are actually needed in this form, given that the review will be part of a special issue on EMT. The paragraph “EMT as a mediator of TIF” is crucial to the review, it should be expanded possibly better explaining the dualism between EMT and epithelial cell plasticity (see Huang S at al Trends Mol Med 2016). Finally, an explanation of the clinical/biological implication of these observations should be included either in the last paragraph “EMT in Hypertensive Neuropathy” or in the conclusions.

R: The paragraphs “Epithelial to Mesenchymal Transition (EMT)” and “Markers of EMT” have been abridged, whereas the paragraph “EMT as Mediator of TIF” has been expanded to better focus the attention of the reader on TIF and hypertension. The concept of plasticity in the kidney has been discussed, and the paper by Huang has been quoted. We thank the Reviewer for these comments and suggestions, which led us to make changes in the text, as suggested.

Reviewer 2 Report

This manuscript well reviews recent progress in the field of EMT in kidney disease.

1. It is well known that renal epithelial cells arise by mesenchymal to epithelial transition (MET) during embryogenesis. Describe it and cite the following references. 

1) EMT-MET in renal disease: should we curb our enthusiasm?

Cancer Lett. 2013 Nov 28;341(1):24-9. doi: 10.1016/j.canlet.2013.04.018. 

2) Transformations between epithelium and mesenchyme: normal, pathological, and experimentally induced.

Am J Kidney Dis. 1995 Oct;26(4):678-90

2. TGF-b directly binds only to type II receptor. Modify the sentence as follows;

“After binding of TGF-b to receptor type II (TbR-II), TGF-b-bound TbR-II phosphorylates type I receptor (TbR-I)……. ”(line 113-114)

Author Response

Referee 2

This manuscript well reviews recent progress in the field of EMT in kidney disease.

R: We thank the Referee for appreciating our work.

1. It is well known that renal epithelial cells arise by mesenchymal to epithelial transition (MET) during embryogenesis. Describe it and cite the following references. 

1) EMT-MET in renal disease: should we curb our enthusiasm?

Cancer Lett. 2013 Nov 28;341(1):24-9. doi: 10.1016/j.canlet.2013.04.018. 

2) Transformations between epithelium and mesenchyme: normal, pathological, and experimentally induced.

Am J Kidney Dis. 1995 Oct;26(4):678-90

R: Thanks for the useful suggestion: we have discussed MET in the revised manuscript and have added these two references.

2. TGF-b directly binds only to type II receptor. Modify the sentence as follows;

“After binding of TGF-b to receptor type II (TbR-II), TGF-b-bound TbR-II phosphorylates type I receptor (TbR-I)……. ”(line 113-114)

R: The sentence has been modified.

Reviewer 3 Report

Authors have thoroughly reviewed the research findings that have been published and that are relevant to the topic. The reviewer has one minor suggestion -

*It would be better if the authors cite recent references.

Author Response

Referee 3

Authors have thoroughly reviewed the research findings that have been published and that are relevant to the topic. The reviewer has one minor suggestion -

*It would be better if the authors cite recent references

R: We thank the Referee for appreciating our review. Following his/her suggestion, we added the following recent references:

1: Hewitson TD, Holt SG, Tan SJ, Wigg B, Samuel CS, Smith ER. Epigenetic Modifications to H3K9 in Renal Tubulointerstitial Cells after Unilateral Ureteric Obstruction and TGF-β1 Stimulation. Front Pharmacol. 2017;8:307. 

2: Liu BC, Tang TT, Lv LL, Lan HY. Renal tubule injury: a driving force toward chronic kidney disease. Kidney Int. 2018;93:568-579. 

3: Hu H, Hu S, Xu S, Gao Y, Zeng F, Shui H. miR-29b regulates Ang II-induced EMT of rat renal tubular epithelial cells via targeting PI3K/AKT signaling pathway. Int J Mol Med. 2018;42:453-460. 

4: Griggs LA, Hassan NT, Malik RS, Griffin BP, Martinez BA, Elmore LW, Lemmon CA. Fibronectin fibrils regulate TGF-β1-induced Epithelial-Mesenchymal Transition. Matrix Biol. 2017;60-61:157-175. 

5: Chen T, You Y, Jiang H, Wang ZZ. Epithelial-mesenchymal transition (EMT): A biological process in the development, stem cell differentiation, and tumorigenesis. J Cell Physiol. 2017;232:3261-3272. 

Round  2

Reviewer 1 Report

My previous comments/suggestions were addressed satisfactorily.